# Aerobic Exercise Inhibits CUMS-Depressed Mice Hippocampal Inflammatory Response via Activating Hippocampal miR-223/TLR4/MyD88-NF-κB Pathway

**DOI:** 10.3390/ijerph17082676

**Published:** 2020-04-14

**Authors:** Honglin Qu, Ruilian Liu, Jiaqin Chen, Lan Zheng, Rui Chen

**Affiliations:** 1College of Physical Education, Yichun University, Yichun 336000, China; quhonglin20040125@126.com (H.Q.); liuruilian5218@126.com (R.L.); 2Key Laboratory of Physical Fitness and Exercise Rehabilitation of Hunan Province, Hunan Normal University, Changsha 410012, China; lanzheng@hunnu.edu.cn (L.Z.); crui1969@163.com (R.C.)

**Keywords:** depression, aerobic training, inflammation in the hippocampus, miR-223, TLR4/MyD88-NF-κB

## Abstract

*Objective*: To investigate the role of aerobic exercise in inhibiting chronic unpredictable mild stress (CUMS) depressed mice hippocampal inflammatory response and its potential mechanisms. *Methods*: Fifty-four male eight-week-old C57BL/6 mice were divided as control group (CG) (18 mice) and model group (36 mice). Model group mice were treated with 13 chronic stimulating factors for 28 days to set up the CUMS depression model. Neurobehavioral assessment was performed after modeling. The mice in the model group were randomly divided into the control model group (MG) and the aerobic exercise group (EG), with 18mice in each group. The EG group carried out the adaptive training of the running platform: 10 m/min, 0° slope, and increased by 10 minutes per day for 6 days. The formal training was carried for 8 weeks with 10 m/min speed, 0° slope, 60 min/d, 6 d/Week. After the training, a neurobehavioral assessment was performed, and hippocampus IL-1β and IL-10 protein levels were detected by ELISA. RT–PCR was used to detect the expression of miR-223 and TLR4, MyD88, and NF-κB in the hippocampus. Western blot was used to detect the expression of TLR4 and phosphorylated NF-κBp65 protein in the hippocampus. *Results*: The hippocampus function of CUMS depression model mice was impaired. The forced swimming and forced tail suspension time were significantly prolonged, and inflammatory factors IL-1β were significantly increased in the hippocampus. Aerobic exercise significantly improves CUMS-depressed mice hippocampal function, effectively reducing depressive behavior and IL-1β levels, and increasing IL-10 levels. Besides, aerobic exercise significantly upregulates the expression level of miR-223 and inhibits the high expression of TLR4, MyD88, and NF-κB. *Conclusion*: Aerobic exercise significantly increases the CUMS-depressed mice hippocampus expression of miR-223, and inhibits the downstream TLR4/MyD88-NF-κB signaling pathway and the hippocampal inflammatory response, which contributes to the improvement of the hippocampal function.

## 1. Introduction

Depression has become a worldwide problem. Patients often experience pleasure and cognitive loss, memory loss, severe high suicidal tendencies, or increased suicidal tendencies, posing a significant burden on patients’ families and society. Recently, studies have shown that the pathogenesis of hippocampal lesions and neuronal cell death in patients with depression might be related to the explosive activation of inflammatory reactions [1]. The “cytokine theory” suggests that stress-stimulated or over-activated immune systems produce inflammatory cytokines and play an important role in the pathogenesis of depression [2], which have been confirmed by clinical trials [3,4,5] and animal experiments [6,7]. Therefore, anti-inflammatory treatment for reducing the hippocampal inflammatory response and improving hippocampal function have become a new target for the prevention and treatment of depression.

Studies have suggested that microRNAs (miRNAs) can directly or indirectly bind to inflammation-related target genes to degrade messenger RNA(mRNA) or prevent mRNA translation from participating in the regulation of inflammatory pathways, and play an important regulatory role in the occurrence and development of inflammatory diseases [8,9,10,11]. Studies showed that the expression of miR-223 was significantly increased during the onset of inflammatory bowel diseases and inflammatory diseases with radiation-induced lung injury, which inhibited the expression of pro-inflammatory factors IL-1β, NLRP3, TNF-α [12], upregulated the anti-inflammatory factor IL-10 express, and reduced the inflammatory response [13,14,15]. MiR-223^-/-^ mice expressed higher levels of NLRP3, leading to increased production of IL-1β and susceptibility to inflammatory responses. Nielsen et al. [16] showed that aerobic exercise could significantly increase blood miR-223 expression levels and improve skeletal muscle function. After a 10 km marathon, the expression level of miR-223 was significantly increased [17]. The authors believed that the inflammatory mechanisms induced by strenuous exercise, such as the circulating inflammatory microRNA-mediated inflammatory cascade, induced miR-223 to counteract the inflammatory response of inflammatory cytokines. Taghizadeh et al. [18] showed that short-term endurance training did not induce significant upregulation of blood hsa-miR-223 in patients with type 2 diabetes, but had a positive effect on platelet function, glycemic index, physical fitness, and body composition in women with type 2 diabetes. However, whether aerobic exercise inhibits hippocampal inflammation and improves hippocampal function in mice with depression by upregulating the expression of hippocampal miR-223 has not been reported. 

It is well known that the TLR4 signaling pathway is involved in the regulation of innate immune responses and adaptive immune responses, and is closely related to the immunoinflammatory process of depression [19], and is a major participant in hippocampal inflammation in chronic unpredictable mild stress depression mice [20]. TLR4 activation through the intracellular IL-1β homologous domain, using the downstream signaling molecule MyD88, mediated the activation of the effectors NF-κB, NF-κB translocated into nuclear phosphorylation and regulated immunoinflammatory factor expression [21,22,23]. The TLR4 signaling pathway recruited and activated downstream IL-1 receptor associated kinase 4 (IRAK4), IRAK1, IRAK2, and TNF-α receptor-associated factor 6 (TRAF6) and activated the NF-κB signaling pathway via the classic MyD88-dependent signaling pathway [24]. It has been confirmed that miR-223 is expressed at a higher level in the differentiation of monocytes into granulocytes, while the mRNA expression levels of TLR4 and TLR3 are lower, suggesting that miR-223 may be negative with TLR4 receptor. Overexpression of miR-223 could reduce differential expression of pro-inflammatory cytokines IL-1β and increase the high expression of anti-inflammatory factor IL-10 in macrophages by inhibiting TLR4 [25,26]. However, whether aerobic exercise regulates the regulation of the TLR4/MyD88-NF-κB signaling pathway through miR-223 has still not been reported. This study is to investigate the effects of aerobic exercise on hippocampal miR-223 and its downstream pathway TLR4/MyD88-NF-κB expression and hippocampal inflammatory response in CUMS-depressed mice.

## 2. Materials and Methods

### 2.1. Instruments and Reagents

Microplate reader (MB16-414, New York, NY, USA), real-time PCR instrument (Bio-Rad CFX96Touch, California, CA, USA), electrophoresis and transfer tank, fluorescence microscope (Nikon Eclipse Ti-SR, Tokyo, Japan), imaging system (Nikon DS -U3, Tokyo, Japan), high-speed tissue homogenizer (T10 Basic, Bartlesville, OK, USA).

Rabbit anti-polyclonal antibody TLR4 (ABclonal, Boston, MA, USA), MyD88 (ABclonal, Boston, MA, USA), NF-κB (ABclonal, Boston, MA, USA), β-actin and goat anti-rabbit secondary antibody (Wuhan Google Biotechnology Co., Ltd. Wuhan, China), DAB chromogenic reagent (DAKO, Glostrup, Denmark), miR-223 primer (Ribobio, Guangzhou, China), TLR4, MyD88, NF-κB (Shanghai Shenggong, Shanghai, China), reverse transcription kit (TAKARA Bio, Los Angeles, USA), mouse IL-1β, IL10 ELISA kit ( ABclonal, Boston, MA, USA).

### 2.2. Animal Processing and Grouping

Briefly, 54 male 8-week-old C57BL/6 mice were obtained from Hunan Slack Jingda Experimental Animal Co., Ltd.(Changsha, China, license number SCXK Xiang 2016–0002). After one week of adaptive feeding, 18 mice were randomly selected as the blank control group (control group, CG). Then, 36 mice were treated with 13 chronic unpredictable stimuli for 28 days to set up the chronic stress-induced (CUMS) depressed model. Neurobehavioral assessment was performed. The remaining mice were randomly divided into the model group (MG) and the aerobic exercise group (EG). The EG group was conducted a moderate-intensity mouse treadmill training for 8 weeks. All animals received humane care in compliance with the experimental protocol approved by the Committee of Laboratory Animals according to institutional guidelines.

### 2.3. CUMS Depression Mouse Model

Thirteen kinds of chronic unpredictable stress stimulating factors were selected, including changes in light properties, day and night adjustment, fasting, water ban, white noise, moist litter, inclined squirrel cage, restraint, high- and low-temperature swimming, horizontal turbulence. The stimulation plan was generated according to the random number method. According to the implementation of different stimuli in the adjacent two days, a fine adjustment was made [27]. One to two kinds of stimuli were treated every day, and the stimulations were continued for 28 days. 

### 2.4. Aerobic Exercise Protocol

The exercise protocol referred to the training program of Bedford et al. [28]. Briefly, the EG group mice were subjected to adaptive training for increasing load in the first week. The starting intensity of the training was 10 m/min, 0° slope, and the training was continued for 6 days according to the daily increment of 10 min. Formal training was performed at 10 m/min speed, 0° slope, 60 min/d, 6 d/week, continuous for 8 weeks.

### 2.5. Neurobehavioral Assessment

After the exercise, 10 mice in each group were randomly selected for neurobehavioral assessment, including forced swimming and forced tail suspension. A diameter of 10 cm, the volume of 2 L round beaker with 10 cm depth of 25 ± 1 °C water, was used for forced swimming. The ANC Core HD1080P HD camera recorded the immobile state within latency 4 min and the last 4 min. The forced tail device was designed with a black bottom, and the top of the box was illuminated by a 25 W incandescent lamp. The immobile state within 6 min and inactive state duration within later 4 min were recorded by V11.60.00 genuine monitoring software in the forced tail test.

### 2.6. Sample Processing

After 8 weeks of aerobic exercise, all mice were fasted overnight. Twelve mice were randomly selected and intraperitoneally injected with 1% sodium pentobarbital (dose: 50 mg/Kg), and the chest was opened rapidly. After the sacrifice, the mice were stripped of the skin on the ice, and the skull was exposed. We opened the skull from the occipital foramen with forceps, fully exposed the brain tissue, carefully peeled off the left and right cortex of the brain. Then, we exposed the whole hippocampus, and peeled off the hippocampus by the glass; we also removed the hippocampus from the cerebral cortex and surrounding brain tissue. In addition, 6 mice hippocampus, which were randomly selected from each group, were hompgenized in an electric homogenizer, and the supernatant was extracted. The protein contents of IL-1β, IL-10, TLR4, and NF-κB were detected according to the ELISA kit procedure. The remaining mice were subjected to brain pre-cooling with 4% paraformaldehyde fixation. Once the mice were stiff and liver completely whitened, the brain tissue were removed from the decapitated ice and fixed in 4% paraformaldehyde for more than 48 hours, after which the tissue was prepared for pathological examination. 

### 2.7. Enzyme-linked Immunosorbent Assay (ELISA)

The blood supernatant was extracted. The content of IL-1β, IL-10, TLR4, TNF-α, and NF-κB was measured according to the ELISA kit procedure from the manufacturer’s instructions. The optical density was measured using a Thermo Scientific Multiskan microplate reader. Each assay was performed in triplicate.

### 2.8. Nissl Staining

Nissl is a basophilic plaque or fine particles. The embedded wax block was cut into 5-μm slices with a fully automatic slicer, routinely dewaxed, and immersed in 1% toluidine blue dye solution for several minutes and then quickly differentiated into 95% alcohol. The xylene was transparent, and the neutral gum was sealed and observed under an optical microscope. In each hippocampal slide, 5 fields of view were randomly selected under high magnification (×200) to observe the morphology of hippocampal CA1 and CA3 neurons, and double-blind counts of pyramidal cells and Nissl bodies were taken with the average value.

### 2.9. Immunohistochemistry Staining

All wax blocks were sectioned to 5-μm thickness by a fully automatic slicer, followed by conventional dewaxing to hydration. The sections were then treated with 3% H_2_O_2_ in distilled water for 10 min and rinsed thoroughly in distilled water, washed in PBS buffer for 5 min before a 10-min incubation with 1% BSA diluted in TBS buffer and repaired with microwave radiation antigen, which was blotted with filter paper. Then, the primary antibodies against IL-1β (1:1000), IL-10 (1:500), TLR4 (1:500), MyD-88 (1:500), TNF-α (1:500), and NF-κB (1:500) were added. After 4 °C overnight incubation, tissues were washed with phosphate-buffered saline Tween-20, and secondary antibodies were added. After incubation at room temperature, diaminobenzidine (DAB) was added to develop colors under 5 min at room temperature, and sections were counterstained with hematoxylin. Then the sections were washed in purified water and mounted by neutral resin. The staining was observed through a microscope (Olympus, Tokyo, Japan), and results were recorded in photos. A positive stain of IL-1β, IL-10, TLR4, MyD-88, TNF-α, and NF-κB were mainly tan, which is located at the cytoplasm. Three fields of view were randomly selected for immunohistochemically staining from the hippocampus of the section, and analyzed by CIIAS98 color pathological image analysis system. The results were gained after the analysis of the ratio between the number of positive cells and the total number of cells.

### 2.10. Association Analysis of miRNAs and mRNA Targeting Regulation

The total RNA of mouse hippocampus was extracted by homogenization of Trizol on ice. After measuring and evaluating the purity and quality by Q5000 Ultramicro spectrophotometer (Quawell, Los Angeles, CA, USA), high-throughput sequencing of mRNA and miRNAs was performed, and the original data of sequencing was determined according to the difference. The upregulation of miRNAs regulates differentially downregulated mRNA and differentially downregulates miRNAs targeting the upregulation of mRNA, and differential miRNAs and differentially expressed mRNA. It also performs functional analysis of GO and KEGG associations of miRNAs targeting regulatory gene sets. The method was reported in Figure 1.

### 2.11. RT–PCR

Total RNA from the hippocampus were extracted with Trizol reagent, and cDNA synthesis was performed using a high-capacity cDNA reverse transcription kit (TAKARA Bio, Tokyo, Japan). PrimeScript® RT Master Mix Perfect Real Time kit (TAKARa Bio, Tokyo, Japan) was used for RT–PCR determination of relative mRNA expression levels of TLR4, MyD88, and NF-κB. GAPDH is a housekeeping gene. The reaction conditions were as follows: 95 °C for 10 min, 1 cycle pre-denaturation; 95 °C for 15 s, 60 °C for 30 s, 65 °C for 30 s, 40 cycles of PCR reaction; 72 °C for 10 min, annealing. Each experiment was replicated three times, and the relative gene expression was calculated by the 2^–ΔΔCt^ method. The base sequence was reported in Table 1.

cDNA was reverse transcribed according to the method of miRNA reverse transcription kit (TAKARA Bio, Tokyo, Japan), and RT–PCR reaction of miRNA was carried out in the same manner as above. The primers were designed and synthesized by Guangzhou RiboBio Biotechnology Co., Ltd., and U6 was an internal reference. The reaction conditions were as follows: 95 °C for 30 s, 1 cycle pre-denaturation; 95 °C for 5 s, 65 °C for 30 s, 39 cycles of PCR reaction, 95 °C for 30 s annealing. The relative gene expression of miR-223 was calculated by 2^–ΔΔCt^ method.

### 2.12. Western Blot Analysis

Mice hippocampus tissue was selected, and 1 ml of pre-cooled tissue lysate was added per 100 mg for tissue lysis. After homogenization on a high-speed tissue homogenizer, it was centrifuged at 12,000× *g* for 5 min at 4 °C. After taking the supernatant, the protein concentration was determined by a microplate reader. After the same amount of protein was loaded, electrophoresed, and transferred. Membranes were blocked in 3% milk-TBST buffer at 4 °C. The membranes were incubated overnight at 4 °C with anti-TLR4 (1:1000), anti-NF-κB (1:1000), and anti-MyD88 (1:1000). The corresponding anti-rabbit IgG secondary antibody (1:2000–5000) was applied for 1 h. The internal reference was the β-actin protein.

### 2.13. Statistics

All experimental data were expressed as mean ± standard deviation ($\bar{X}$ ± SD). ANOVA analysis of variance was performed on the experimental data obtained by SPSS20.0 statistical software (IBM, Vancouver, British Columbia, Canada). For differences in frequency between the two groups, the chi-square test was used. If the data are not normally distributed, the difference in multiple groups was analyzed by the Kruskal–Wallis test. Significant difference at each comparison point was set at *p* < 0.05.

## 3. Results

### 3.1. Neurobehavioral Assessment 

The results of neurobehavioral assessment showed that the forced time of forced swimming and forced tail suspension were significantly longer in the MG than in the CG (*p* < 0.01, *n* = 10). Compared with the MG, the immobility time was significantly decreased in the EG group (*p* < 0.01; Figure 2). The results showed that moderate-intensity aerobic exercise could alleviate the desperate behavior of depressed mice.

### 3.2. The Changes of Nissle Body in the Hippocampus

The results of Nissl staining showed that the pyramidal cells in the hippocampal CA1 region of the mice were neatly arranged, and the number of pyramidal cells was not significantly different. In CG, hippocampal CA3 depyramidated cells were neatly arranged, the cytoplasm was full, and the Nissl bodies were clear. The pyramidal cells in the hippocampal CA3 region of model mice were sparsely arranged and disordered with obvious nuclear shrinkage. The cytoplasm was lightly colored and showed significant dissolution or disappearance of Nissl bodies. Compared with MG, the pyramidal cells in the hippocampal CA3 region of the EG were arranged neatly, the number of Nissl bodies was increased, the phenomenon of nuclear shrinkage was decreased, and the cytoplasm content was increased. The result of Nissl staining was reported in Figure 3 and Table 2.

### 3.3. Correlation Analysis of Targeting the TLR4/MyD88-NF-κB Pathway by miR-223 Activation Based on Aerobic Exercise

Correlation analysis of high-throughput sequencing results showed that in the hippocampus, aerobic exercise induced miR-223 downregulated TLR4 expression through negative regulation and targeted regulation of the TLR4/MyD-88-NF-κB signaling pathway expression directly affects the expression of the inflammatory cytokines IL-1β and TNF-α. The correlation analysis was reported in Figure 4.

### 3.4. The Immunohistochemistry 

The results of immunohistochemistry showed that the inflammatory cytokines IL-1β, TNF-α, and TLR4/MyD88-NF-κB signaling pathway-related factors in the hippocampus CA1 and CA3 regions of MG mice were significantly increased in expression levels compared with CG (*p* < 0.01). The expression of IL-10 in the hippocampal CA1 region of MG mice also increased slightly; however, there was no significant difference. The CA3 region increased significantly (*p* < 0.05). Compared with MG, the expression levels of inflammatory cytokines IL-1β and NF-κB in the hippocampal CA1 region of EG mice were significantly downregulated (*p* < 0.01), and TLR4, MyD88, and TNF-α were significantly reduced (*p* < 0.05). Each of the pro-inflammatory factors in the CA3 region decreased significantly compared with the CA1 region (*p* < 0.01). The expression of IL-10 showed a very significant increase (*p* < 0.01). This suggested that aerobic exercise had a more effective intervention effect in the CA3 region than the CA1. The results of immunohistochemistry was reported in Table 3 and Figure 5.

### 3.5. ELISA

The results of the test showed that the TLR4, TNF-α, and IL-10 in the hippocampus of the MG were significantly increased (*p* < 0.01), and IL-1β and NF-κB were increased, compared with CG mice. However, compared with MG, the content of TLR4 and NF-κB in the hippocampus of EG showed a significant decrease (*p* < 0.01), IL-1β and TNF-α were decreased, and IL-10 was increased very significantly (*p* < 0.01). The results also displayed that the content of NF-κB in the hippocampus of EG was significantly lower than that of CG, which might be closely related to the intervention of systematic aerobic exercise. The results of the test was reported in Table 4.

### 3.6. Hippocampal miR-223 Expression

RT–PCR showed that compared with the CG group, the expression of miR-223 in the hippocampus of MG mice was significantly increased (*p* < 0.01, *n* = 6). Compared with MG, miR-223 expression of the EG mice was significantly increased, which indicates that aerobic exercise could upregulate miR-223 expression in the hippocampus. The result was reported in Figure 6.

### 3.7. RT–PCR

The results of RT–PCR showed that the pro-inflammatory factors IL-1β and TLR4 of the MG hippocampus were significantly upregulated (*p* < 0.05), compared with CG, while MyD88 and NF-kB were very significantly up-regulated (*p* < 0.01). There was no significant difference in the anti-inflammatory related factor IL-10. However, the expression of TLR4 and MyD88 of EG mice were significantly downregulated compared with MG (*p* < 0.01), NF-κB was significantly downregulated (*p* < 0.05), and IL-10 was upregulated significantly (*p* < 0.01). Unexpectedly, although IL-1β was downregulated, it was not significantly different from MG. The results of RT-PCR was reported in Table 5.

### 3.8. Western Blot

The results of Western blot showed that compared with CG, the expression levels of TLR4, MyD88, and NF-κB in MG mice hippocampus were significantly increased. Compared with MG, the expression levels of TLR4 and MyD88 in EG mice hippocampus were significantly reduced respectively (*p* < 0.01). The expression of NF-κB was significantly reduced (*p* < 0.05), as shown in Figure 7.

## 4. Discussion

In recent years, aerobic exercise has been reported to antagonize inflammatory responses to protect brain tissue from damage. Studies have confirmed that aerobic exercise could effectively improve neuronal damage and reduce neuronal loss in the cerebral cortex of mice with chronic cerebral ischemia, play a protecting role in the cranial nerves. The expression of Jagged1 and Ncx1 mRNA in brain tissue and peripheral blood was highly correlated with exercise-induced neuroprotection [29]. Recent studies have also found that aerobic exercise participates in the regulation of the hypothalamic–pituitary–adrenal axis and related inflammatory factors to improve the body’s inflammation in mice with brain damage [30]. Kohut et al. [31] showed that exercise could exert antidepressant effects by reducing the pro-inflammatory factors IL-6 and IL-18, C-reactive protein, and TNF-α. Liu et al. [32] showed that the expression of IGF1, ERK, and CREB was increased in experimental mice through a 4-week aerobic treadmill exercise. The results of this study showed that the despair behavior of FST and TST in the MG group was significantly enhanced, and the hippocampus pro-inflammatory factor IL-1β level was significantly increased. Aerobic exercise could significantly reduce the desperate behavior of mice and effectively reduce IL-1β levels and increase the expression level of the anti-inflammatory factor IL-10. It was speculated that aerobic exercise could reduce the inflammatory injury caused by CUMS depression while improving the depression of mice. 

It has been reported that miR-223, as a kind of miRNA specifically expressed in myeloid cells, is a novel regulator of the inflammatory response [33] and a negative regulatory effector of inflammatory cytokines such as TLR4 [34,35,36]. Reports showed that miR-223 might be a “promoter” that transitions from a pro-inflammatory process to an anti-inflammatory process [37,38]. Studies have found that miR-223 and the inflammatory factors TNF-α, IL-1β, and IL-6 were highly expressed in the lesions of acute spinal cord injury in mice, which reached the maximum peak at 12–24 h after injury [39,40]. There was a positive correlation between the level of miR-223 and the inflammatory factor IL-6 in the acute phase of injury, which indicates that miR-223 might be involved in the regulation of inflammatory factor-mediated spinal inflammatory injury, and its anti-inflammatory effect might be related to the severity and duration of disease damage.

Recent studies have found that miR-223 is an important mediator of neuronal development, prominent plasticity, and neurodegeneration in the normal hippocampus [41], and is highly expressed in the hippocampus, especially after hippocampal injury [42,43]. Our study shows that hippocampus miR-223, the expression levels of TLR4, MyD88, and NF-κBp65 in the hippocampus, were significantly higher in MG mice. It reveals that miR-223 might be involved in the pathogenesis of depression hippocampal inflammation. Aerobic exercise could significantly increase the content of miR-223, reduce the expression of TLR4, MyD88, and NF-κBp65 in the hippocampus, reduce the content of IL-1β in the serum, and significantly increase the expression of IL-10. This might be related to miR-223 participating in negative feedback to regulate depression and inflammation. The specific mechanism of its involvement in inflammatory response might be related to the level of miR-223 expression. Although it has been confirmed that miR-223 plays an important regulatory role in the inflammatory response, there are few reports on the way in which miR-223 is used in disease to switch from pro-inflammatory response to anti-inflammatory response, and a lack of exact mechanism research. We speculate that the process of miR-223 negatively regulating inflammatory response might be related to the level of expression. However, we did not carry out miR-223 gene interference to explore the specific mechanism of its targeted regulation of the inflammatory response process because of insufficient funding. Some studies [44] have pointed out that in the Toll-like receptor-induced inflammatory response, downregulation of miR-223 could induce the activation of NF-κB and MAPK signals, which promote TNF-α, IL-6, and IL-1β production during LPS stimulation. Similar studies have also suggested that the deletion of miR-223 was likely to aggravate ethanol-induced liver injury, neutrophil infiltration, and upregulation of IL-6 expression [45]. Previous studies have confirmed [46] that in stroke or excitotoxic neuronal disease, miR-223 could inhibit neuronal damage by regulating the functional expression of glutamate receptor subunits Glur2 and NF2B in the brain. However, the lack of miR-223 easily leads to high expression of hippocampal neurons NR2B and Glur2 and induces neuronal damage. A recent study [47] found that miR-223 increased the expression of primary TLR4 and STAT3, as well as the expression of TLR4, STAT3, and NOS2 in LPS-stimulated primary macrophages, and the high expression of miR-223 Negative regulation of the role of the TLR4/FBXW7 axis reduced the damage of macrophage inflammatory effects on adipose tissue.

Sari et al. [48] showed that the expression of MyD88 and NF-κBp65 protein was increased in LPS-treated rats, which might be determined by miR-223. Recent studies [49,50] have found that miR-223 could slow tissue damage by inhibiting inflammatory responses. As mentioned earlier, Taghizadeh’s studies reported that short-term aerobic exercise could increase the expression of platelet miR-223 in women with type 2 diabetes. De found that the specific circulating inflammatory response and miR-223 were increased in acute high-intensity aerobic exercise, where the higher expression of miR-223 contributed to resist inflammation induced by acute high-intensity exercise. Our study found that aerobic exercise could upregulate the expression of miR-223 in the hippocampus, inhibit the high expression of TLR4, MyD88, and NF-κB, decrease the expression of inflammatory factor IL-1β, and enhance the expression of anti-inflammatory factor IL-10. Thus, we hypothesize that aerobic exercise could increase the expression of miR-223 in the hippocampus of CUMS-depressed mice and slow down the hippocampal inflammatory response in depressed mice, which is mediated by the activation of the TLR4/MyD88-NF-κB pathway. 

TLR4 could induce differential expression of MyD88-NF-κB dependent and independent inflammatory signaling pathways, directly inducing the occurrence and development of inflammatory responses [51]. It has not been reported whether there is a differential expression of miR-223 in the negative feedback intervention of signal pathway factors such as TLR4, especially whether this mechanism of action is involved in the process of the inflammatory response in the hippocampus of depression. The previous study found that the expression of miR-223 was enhanced after TLR4 was inhibited. This was in contrast to Yan et al. [52], where downregulation of miR-223 induced the upregulation of TLR4, NF-κB, and NLRP3, while inhibiting TLR4 in vitro reduced the release of in vitro inflammatory factors following the downregulation of miR-223. This confirmed that miR-223 targets the role of TLR4 in the inflammatory response. Our results demonstrate that miR-223 negatively regulates the activation of the TLR4/MyD88-NF-κB pathway, consistent with the activation in Wang’s 2015 study, which also demonstrated that miR-223 silencing enhanced LPS-induced expression of LTR4 and its downstream signaling factor NF-κB and potentiated the inflammatory response. miR-223 directly affects the expression levels of inflammatory factors IL-1β and TNF-α by regulating the TLR4/MyD88-NF-kB signaling pathway(Figure 8). This result was also confirmed by Wu’s 2019 study. In addition, miR-223 could also inhibit the expression of inflammatory factors such as IL-6 by targeting NLRP3 [53], upregulate the expression of IL-6 in synoviocytes by downregulating the expression of IL-17 receptor D [54], and, through the IGF-1/PI3K signaling pathway, affect the secretion of IL-6 in mast cells [55]. 

This study demonstrates that aerobic exercise can induce the expression of miR-223, negatively regulate the activation of the TLR4/MyD88-NF-κB signaling pathway, and participate in the inflammatory process of the hippocampus in CUMS-depressed mice. Among them, miR-223 can be used as an effective positive regulator of inflammation and play a role in antagonizing the inflammatory response of depressed hippocampus.

## 5. Conclusions

Aerobic exercise can significantly increase miR-223 expression in CUMS-depressed mice, activate the miR-223/TLR4/Myd88-NF-κB pathway, slow down hippocampal tissue inflammation, and improve hippocampal function in CUMS-depressed mice. This study confirms that miR-223 can downregulate the activation of the TLR4/Myd88-NF-κB pathway, reduce the expression level of inflammatory cytokines, and antagonize the inflammatory process in the hippocampus of depressed mice. The study of aerobic exercise intervention miR-223 expression and protective effects on the hippocampus and other organs would provide new targets for the treatment and rehabilitation of patients with depression and cranial nerve diseases. However, we only confirmed the mechanism involved in the miR-223-targeting TLR4/Myd88-NF-κB signal transduction pathway through the association analysis of miRNAs and mRNA high-throughput sequencing combined with cellular molecular biology. The negative feedback mechanism of this regulation was not verified by miR-223 or TLR4/Myd88-NF-κB signaling pathway blockade or knockdown or other gene interference means. The next study will focus on the verification of the mechanism of feedback regulation between miR-223 and this signaling pathway and its gene interference. In addition, it also preexisted from the cell culture mode through the in vitro negative feedback mechanism to verify the role of the mechanism. 

## Figures and Tables

**Figure 1 ijerph-17-02676-f001:**
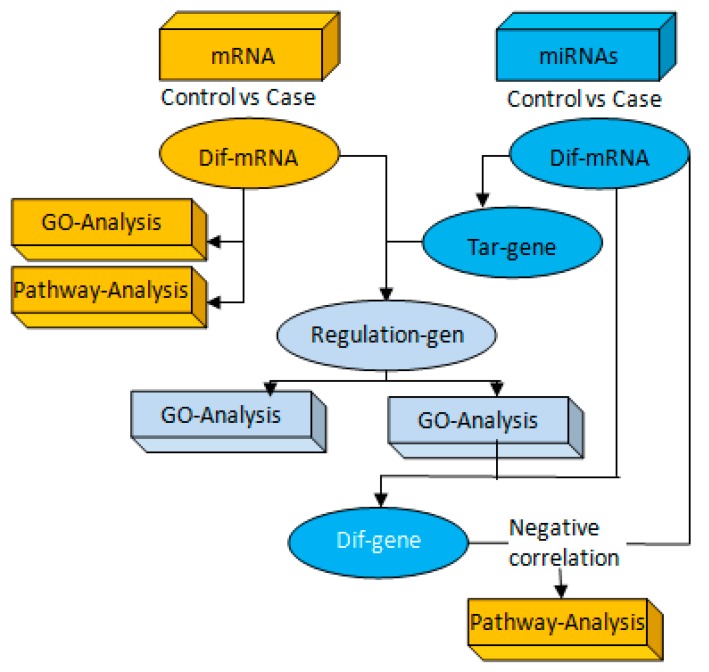
Combined analysis of messenger RNA(mRNA) and microRNAs (miRNAs). Firstly, we set up miRNAs and mRNA sequencing analysis of several groups with the same comparative combination to conduct miRNA–mRNA association analysis. Then, if the transcriptome analysis of the group comparison results in differential genes, and the miRNAs analysis results in differential miRNAs and corresponding target genes, the integration analysis of miRNAs and mRNA could be performed for the group comparison.

**Figure 2 ijerph-17-02676-f002:**
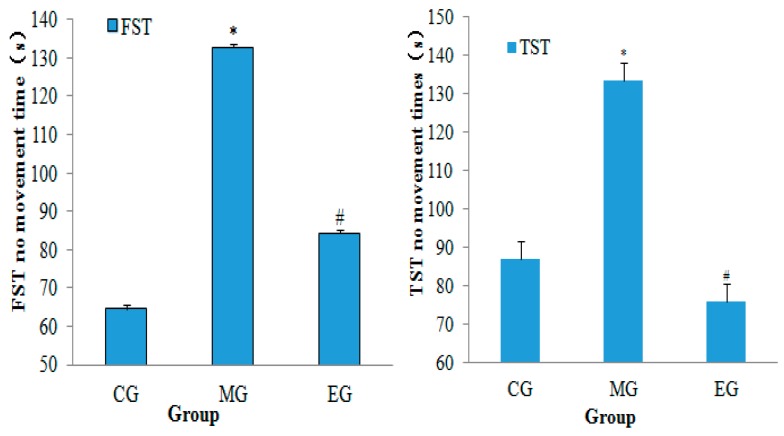
The changes of the total time of forced swimming and forced tail suspension. After exercise, each group consisted of 10 animals. The data were presented as the means ± SD. * *p* < 0.01, versus control group, # *p* < 0.01, versus model group (one-way ANOVA followed by the Kruskal–Wallis test).

**Figure 3 ijerph-17-02676-f003:**
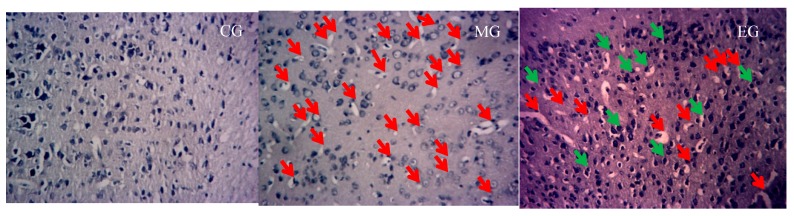
The Nissl staining results of the hippocampus in each group of mice under high magnification (×200). The first picture represents the CA3 of the control group of mice. The second represents model mice. The third represents the exercise mice. In the picture, red represents Nissl pyknosis, green for restored Nissl.

**Figure 4 ijerph-17-02676-f004:**
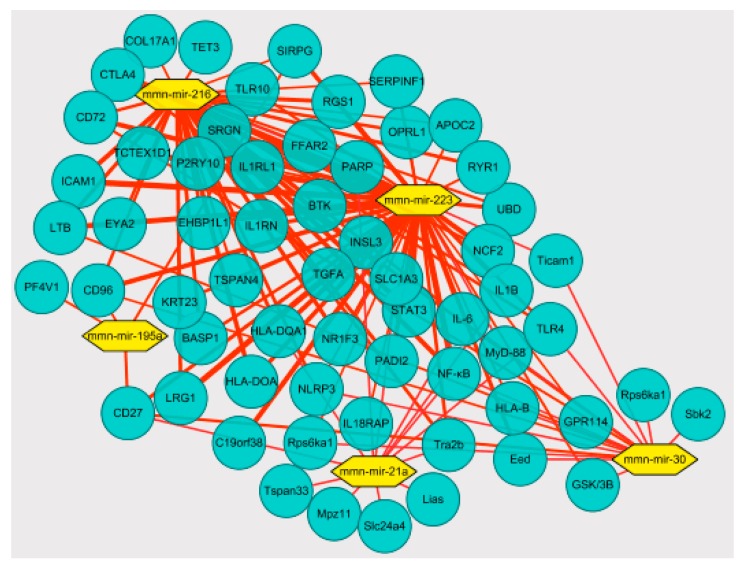
The miR-223 targeted regulation of the TLR4/MyD-88-NF-κB map. According to the data file of the targeting relationship between differentially expressed miRNAs and mRNA, Cytoscape software was imported for visual operation, and the relationship between mRNA and miRNAs was produced. The network included 5miRNAs and multiple mRNAs, including genes related to inflammation. The blue nodes represent mRNA; the yellow nodes represent miRNAs.

**Figure 5 ijerph-17-02676-f005:**
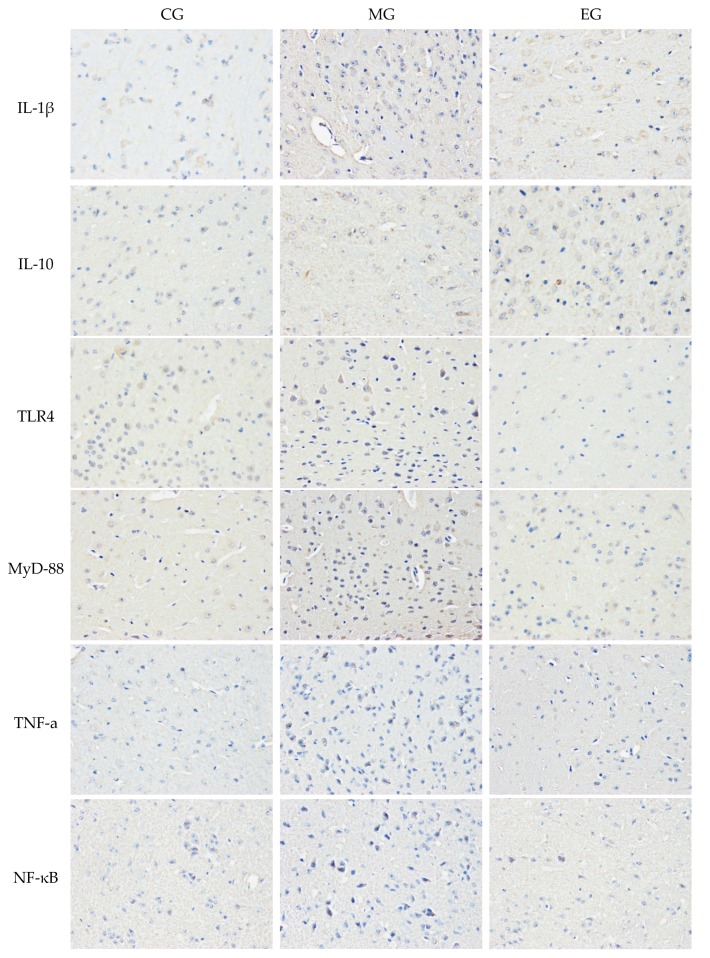
The immunohistochemical detection results of inflammation-related factors in the hippocampus of CUMS mice after 8 weeks of exercise (original magnification 400×). There were six mice for immunohistochemistry analysis in each group. Three fields of view were randomly selected for immunohistochemically staining from the hippocampus of the section, and analyzed by CIIAS98 color pathological image analysis system.

**Figure 6 ijerph-17-02676-f006:**
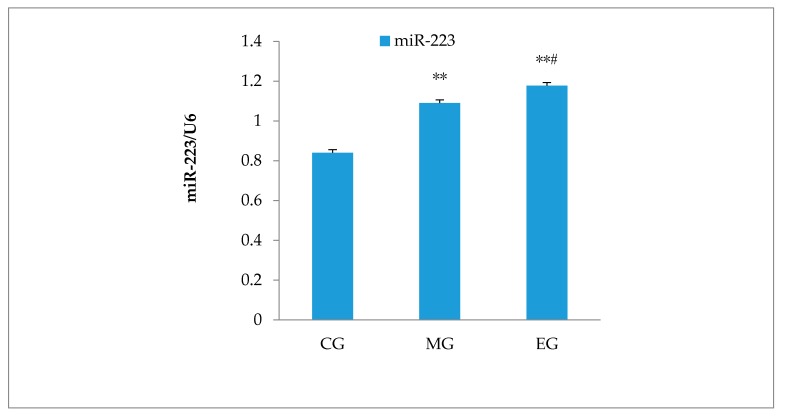
The miR-223 expression in the hippocampus of CUMS mice after 8 weeks of exercise by the RT–PCR assay. The data were presented as means ± SD; there were six mice in each group. U6 was set as the reference gene. ** *p* < 0.01, versus control group, ^#^
*p* < 0.05, versus model group.

**Figure 7 ijerph-17-02676-f007:**
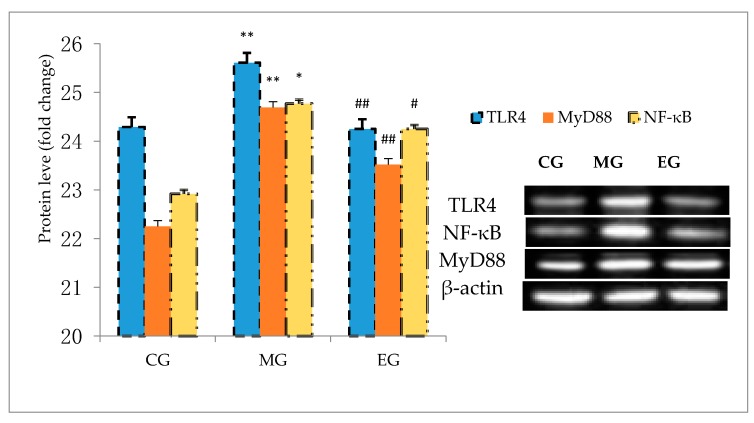
Changes of TLR4, MyD88, and NF-κBp65 expression of the hippocampus of each group. The results of expression of the TLR4, MyD88, and NF-κB65 of the mice of the 3 goups are presented as the means ± SD. The internal reference was the β-actin protein. ** *p* < 0.01, * *p* < 0.05, versus control group, ## *p* < 0.01, # *p* < 0.05, versus model group (one-way ANOVA followed by the Kruskal–Wallis test).

**Figure 8 ijerph-17-02676-f008:**
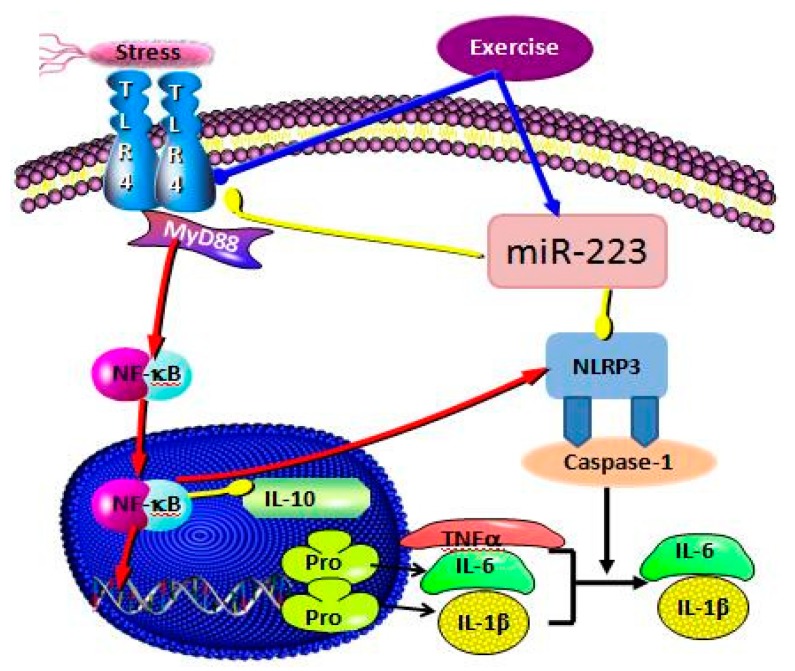
The activation mechanism of miR-223 negative regulation of the TLR4/MyD88-NF-κB signaling pathway.

**Table 1 ijerph-17-02676-t001:** The primer sequences of gene.

Gene	Upstream Primer	Downstream Primer
TLR4	5′-ACAAACGCCGGAACTTTTCG-3′	5′- GTCGGACACACACAACTTAAGC-3′
MyD88	5′-CGCCGCCTATCGCTGTTCTTG-3′	5′- TTCTCGGACTCCTGGTTCTGCTG-3′
NF-κB	5′-ATCATCGAACAGCCGAAGCA-3′	5′-TGATGGTGGGGTGTGTCTTG-3′
GAPDH	5′- CATGGCCTTCCGTGTTCCTA-3′	5′- CCTGCTTCACCACCTTCTTGAT-3′

**Table 2 ijerph-17-02676-t002:** Effects of aerobic exercise on the number of pyramidal cells and intact Nissl bodies in the hippocampal CA1 and CA3 regions of depressed mice (*n* = 6).

Group	CA1 (Number)	CA3 (Number)	Nissl Bodies (Number)
CG	56.8 ± 6.6	31.3 ± 8.6	81.3 ± 15.6
MG	55.5 ± 4.9	22.7 ± 9.3 **	51.0 ± 17.9 **
EG	55.2 ± 5.8	29.6 ± 7.0 ^##^	72.6 ± 14.5 ^#^

Note: ** *p* < 0.01, VS CG; ^##^
*p* < 0.01, ^#^
*p* < 0.05, VS MG.

**Table 3 ijerph-17-02676-t003:** Effects of aerobic exercise on the expression of inflammation-related factors in hippocampal CA1 and CA3 regions of depressed mice (*n* = 6).

Group	CG-CA1	CG-CA3	MG-CA1	MG-CA3	EG-CA1	CG-CA3
IL-1β	31.64 ± 5.79	25.13 ± 2.87	51.36 ± 7.12 **	52.04 ± 5.99 **	30.07 ± 3.55 ^##^	32.17 ± 2.30 ^##^
IL-10	17.37 ± 2.45	18.81 ± 2.76	20.62 ± 3.75	22.19 ± 2.75 *	38.67 ± 5.31 ^##^	41.11 ± 5.56 ^##^
TLR4	27.13 ± 4.55	26.37 ± 6.19	45.92 ± 6.24 **	47.90 ± 7.33 **	38.29 ± 7.54 ^#^	33.03 ± 6.71 ^##^
MyD88	22.81 ± 4.77	28.39 ± 6.30	39.12 ± 7.76 **	42.19 ± 7.23 **	35.67 ± 5.31 ^#^	32.69 ± 7.58 ^##^
TNF-α	15.49 ± 3.25	14.04 ± 1.37	31.99 ± 3.06 **	36.55 ± 2.79 **	24.52 ± 3.37 ^#^	21.93 ± 3.40 ^##^
NF-κB	22.71 ± 4.54	20.33 ± 3.06	47.07 ± 6.39 **	50.92 ± 4.98 **	33.84 ± 6.71 ^##^	26.40 ± 3.91 ^##^

Note: ** *p* < 0.01,* *p* < 0.05, VS CG; ^##^
*p* < 0.01, ^#^
*p* < 0.05, VS MG.

**Table 4 ijerph-17-02676-t004:** Changes of expressions of IL-1β and IL-10 in the hippocampus of mice.

Group	IL-1β (pg/mL)	IL-10 (pg/mL)	TLR4 (ng/mL)	NF-κB (pg/mL)	TNF-α (pg/mL)
CG	1.71 ± 0.90	5.39 ± 2.67	24,140.00 ± 1560.39	19.99 ± 1.29	4.90 ± 0.51
MG	4.55 ± 1.37 *	7.36 ± 2.93 **	34,720.00 ± 1436.20 **	29.98 ± 1.42 **	8.78 ± 0.46 **
EG	2.71 ± 1.69 ^#^	13.68 ± 5.74 ^##^	24,845.00 ± 1234.14 ^##^	13.98 ± 1.34 ^*##^	5.68 ± 0.63 ^#^

Note: ** *p* < 0.01, * *p* < 0.05, VS CG; ^##^
*p* < 0.01, ^#^
*p* < 0.05, VS MG.

**Table 5 ijerph-17-02676-t005:** RT–PCR results of inflammation-related factors in mouse hippocampus.

Group	IL-1β	IL-10	TLR4	MyD88	NF-κB
CG	1.24 ± 0.042	1.28 ± 0.040	1.14 ± 0.033	1.16 ± 0.062	1.11 ± 0.041
MG	1.29 ± 0.033 *	1.33 ± 0.071	1.19 ± 0.028 *	1.61 ± 0.096 **	1.20 ± 0.034 **
EG	1.26 ± 0.058	1.42 ± 0.004 **^#^	1.16 ± 0.035 ^##^	1.39 ± 0.033 ^##^	1.12 ± 0.041 ^#^

** *p* < 0.01, * *p* < 0.05, VS CG; ^##^
*p* < 0.01, ^#^
*p* < 0.05, VS MG.

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
