# Peer review of "Aerobic Exercise Inhibits CUMS-Depressed Mice Hippocampal Inflammatory Response via Activating Hippocampal miR-223/TLR4/MyD88-NF-κB Pathway"

_ijerph, 2020, doi:10.3390/ijerph17082676_

Round 1

Reviewer 1 Report

Comments to the author

In this study, Honglin QU, et al. investigated the anti-inflammatory effect of aerobic exercise in CUMS-depressed mice hippocampus, focusing on the change of miR-223 expression. They discovered that aerobic exercise improves the depressive behavior and hippocampal inflammation which were induced by CUMS. Interestingly, although CUMS resulted in an increase in the expression of miR-223 in hippocampus, aerobic exercise induced a further increase. MiR-223 is known as regulator of negative feedback against inflammation. Based on the association analysis of miRNAs-mRNA targeting regulation, the authors claim that exercise-induced miR-223 expression reduced inflammation in hippocampus. Overall, individual phenomenon which are reported in the manuscript are interesting and may indicate the potential mechanism underlying anti-inflammation effect of exercise in CUMS-induced depression, while relationship among them is slightly vague. Therefore, some additional analysis or discussion is necessary.

Major comments

  1. While the authors claim that the increase in exercise induced-miR-223 expression is one of the mechanisms of anti-inflammation effect by exercise based on the association analysis of miRNAs-mRNA targeting regulation, miR-223 was significantly increased by even only CUMS compared to CG in figure 5. Please discuss this discrepancy. For example, do the authors think that significant increase of miR-223 after CUMS did not affect inflammatory condition in CUMS-depressed mice? Or is the increase of miR-223 expression by CUMS compensatory response?

  1. The authors should show more detailed experimental condition and data of association analysis of miRNAs-mRNA targeting regulation including data of MG group.

  1. It would be interesting if the increased expression of miR-223 by exercise overlapped with less expression of pro-inflammatory regulators. Is it possible to show by in situ hybridization of miR-223.

  1. Page 6, line 1; In terms of the change in number of Nissl bodies and nuclear pyknosis, please quantify. Also, what is the criteria of restored Nissl?

  1. Fig.4; It seems to me that the difference between CG and MG, or MG and EG is just the difference of background staining. The authors should show which cells (area) are (is) stained, and quantify them.

  1. Page 9, line 5 from the bottom and figure 6; Although the authors described the change in the phosphorylation of TLR4, MyD88 and NF-κB in the text, the figure legend does not say so. Please describe clearly. Again, the authors need to show quantified data (p-values were shown in the text, but no graphs). The information about antibodies against phosphorylated-form proteins is missing in the “material and methods”.

Specific comments

  1. Paired t test is not suitable for statistical analysis in this experiment. The author should use more ideal statistical analysis as non-paired post-hoc analysis after ANOVA.

  1. Page 9, 3.7. RT-PCR; Although the authors described that “The results of RT-PCR showed that the pro-inflammatory factors IL-1β and TLR4 of MG hippocampus were significantly up-regulated, compared with CG, while MyD88 and NF-kB were very significantly up-regulated, which was different from the ELISA results”, the data is compatible with ELISA data.

  1. There are some writing errors in the manuscript.

Page 2, line 6 from the bottom; "CUMS-deficient" is probably "CUMS-depressed".

Page 2, line 11, Page 3, line 10 and Page 6, line 2 from the bottom, there is unnecessary capital letter.

Page 2, line 17 from the bottom, there is a grammatical error. The manuscript should be proofread again.

Reviewer 2 Report

In this manuscript, the authors investigated the effect of aerobic exercise on hippocampal inflammatory response in CUMS-depressed mice. They suggested that aerobic exercise can inhibit the hippocampal inflammatory response by increasing the miR-233 expression in hippocampus and inhibiting the downstream TLR4/MyD88-NF-kB. This is an interesting study that could enhance our understanding of the crucial role played by miR-233 in inflammatory diseases. However, this manuscript is poorly written and needs extensive revising.

  1. In introduction, page 2, (Studies showed that the expression of miR-223 is significantly increased in the onset of inflammatory diseases, inhibiting the expression of pro-inflammatory factors IL-1β, NLRP3, TNF-a, etc., up-regulating the anti-inflammatory factor IL-10 Express, reduce the inflammatory response) , this paragraph needs to be restructured, add connectives to make it much more understandable.

  1. The methods are poorly written, especially sample processing, immunohistochemistry staining, and western blot. For example, in page 4, in immunohistochemistry staining, they wrote (distilled water wash (2min x 3 times), PBS rinse(2min x 3 times), microwave radiation antigen repair, PBS rinse(2min x 3 times), filter paper suction after the PBS solution around the dry tissue), the methods should be written in passive, the sentences were not linked using any connectives and were not clearly described. In ELISA, although they used a kit, a brief description should be added to explain how they used it.

  1. The authors did not refer to most figures in the main text. Also, all figure legends were short, more details should be added for all of them.

  1. In figure 2, the labels are not clear.

  1. Figure 6, it is not well labelled, I can not see which one is MG, CG or EG.

  1. In page 10, in the discussion, references should be added to (Studies have confirmed that aerobic exercise can effectively improve neuronal damage and reduce neuronal loss in the cerebral cortex of mice with chronic cerebral ischemia, playing a protecting role in the cranial nerves. )

 Typing error:

  1. Page 1, in methods, (Fifty four 8 weeks old male), rewrite this sentence, you could write fifty four male 8 week-old mice.
  2. Page 1, in methods and in different places as well in the manuscript, be consistent how to write the units, for instance, either write 10m/min or 10 m/min, just leave space between the number and the unit.
  3. Page 1, in results, Aerobic, A should be written in lowercase.
  4. Page 1, in keywords, Training, T should be written in lowercase.
  5. Page 2, has-miR-233, should be hsa-miR-233.
  6. All in vitro and in vivo should be Italics.
  7. Page 2, Factor, F should be written in lowercase.
  8. Page 3, in animal processing and grouping, remove the second male from ( 54 male 8-weeks old male).
  9. Page 3, in neurobehavioral assessment, (10cm depth f 25±1) what does the f stand for?
  10. Page 3, in sample processing, 6 mice hippocampus, m should be written in capital.
  11. Page 6, pathway expression Directly affect, D should be written in lowercase.

Reviewer 3 Report

The quality of the text and data presentation is very poor.

the manuscript submitted by Honling Qu et collegues needs an extensive editing of the english language (i.e Grammar, sentence construction,  punctuation, space between words).

In the text there are also several technical errors and non sense sentences

  • "...standard gingival animal dry feed and free diet" (what is this ??)
  • "the ophtalmologist (what ??)... peeled off the hippocampus by the glass"...
  • "With the cerebral cortex and surrounding brain tissue, remove the hippocampus, rinse with ice PBS solution, filter paper to absorb excess water, weigh, 6 left hippocampus of them were soaked in Trizol, frozen at -80 °C, extract protein for RT-PCR detection (protein ??), and the 6 right hippocampus of the mice were frozen in liquid nitrogen and frozen in a -80 °C refrigerator for western blot analysis.
  • "...3% of (W/V) skim milk was blocked by TBST for 2 h"

Moreover, they talk about an high-throughput sequencing of mRNA and miRNAs "....Up-regulation of miRNAs regulates differentially down-regulated mRNA, differentially down-regulates miRNAs targeting up=regulation of mRNA, and differential miRNAs and differentially expressed mRNA, and performs functional analysis of GO and KEGG associations of miRNAs targeting regulatory gene sets"...but did not show any difference between the groups...instead they show only something that looks like an interaction map (fig. 3).

Further they write:

"...The results of western blot showed that compared with CG group, the phosphorylation levels of TLR4, MyD88 and NF-κB in MG group hippocampus were significantly increased. Compared with MG group, the phosphorylation levels of TLR4 (p<0.01) and MyD88(p<0.05) in EG group hippocampus were significantly reduced, respectively. Although NF-κB did not show significant differences(p>0.05), but there is still a certain extent of decline, as shown in Figure 6."

But in the methods they declared that used"... anti-TLR4 (1:1000), anti- p-NF-κBp65 (1:1000)"

So apparently with these antibody they can detect only the phosphorilated NF-kB and they did not compared it the normal NF-kB. 

Author Response

Dear Reviewers:

   Thank you for your letter and for the reviewers’ comments concerning our manuscript entitled “Aerobic exercise inhibit CUMS-depressed mice hippocampal inflammatory response via activating hippocampal miR-223/TLR4/MyD88-NF-κB pathway” (ID: ijerph-740055). Those comments are all valuable and very helpful for revising and improving our paper, as well as the important guiding significance to our researches. We have studied comments carefully and have made correction which we hope meet with approval. We have substantially revised manuscript after reading the comments provided by the three reviewers. Revised portion are marked in yellow in the paper. Please see the attachment.                  

 Special thanks to you for your good comments.

                                                          Yours sincerely,

                                                          Honglin QU

Round 2

Reviewer 1 Report

While the quantitative data has been added, Fig.5 and Fig.7 are still not convincing. Expression does not appear to be reduced in EG, especially for TLR4 in immunohistochemistry and NfKB in Western blots.

Author Response

Dear Reviewers:

   Thank you for your letter and for the reviewers’ comments concerning our manuscript entitled “Aerobic exercise inhibit CUMS-depressed mice hippocampal inflammatory response via activating hippocampal miR-223/TLR4/MyD88-NF-κB pathway” (ID: ijerph-740055). Those comments are all valuable and very helpful for revising and improving our paper, as well as the important guiding significance to our researches. We have studied comments carefully and have made correction which we hope meet with approval. Revised portion are marked in yellow in The paper. The main corrections in the paper and the responds to the reviewer’s comments are as flowing:

   Thanks you very much for your comments and suggestions.

Major Comments:

   While the quantitative data has been added, Fig.5 and Fig.7 are still not convincing. Expression does not appear to be reduced in EG, especially for TLR4 in immunohistochemistry and NfKB in Western blots.

   Answer: It is really true as your suggested that Fig 5 and Fig 7 were still not convincing and expression did not appear to be reduced in EG. We are very sorry for our negligence of this picture. This is our problem, because we selected 6mice of each group for immunohistochemistry and western blot detection, and for each slice of immunohistochemistry, 3 fields of vision were screened. We did not notice this problem at that time, and randomly selected the immunohistochemistry chart of the corresponding group. We have replaced the corresponding pictures.

   Special thanks to you for your good comments.

                                                      Yours sincerely,

                                                          Honglin QU

Reviewer 2 Report

The authors have addressed most of my comments, however, there are few more corrections need to be sorted:

Page 4, distilled water wash (2min x 3times), this sentence need to be rewritten, and in the same paragraph ( dropwise at 37°C30min), add for between C and 30min.

Page 5, diaminobenzidine (DAB) were added to develop colors under 5min at room) replace under with for 5min.

Also, all figure legends were short, more details should be added for all of them.

Author Response

Dear Reviewers:

   Thank you for your letter and for the reviewers’ comments concerning our manuscript entitled “Aerobic exercise inhibit CUMS-depressed mice hippocampal inflammatory response via activating hippocampal miR-223/TLR4/MyD88-NF-κB pathway” (ID: ijerph-740055). Those comments are all valuable and very helpful for revising and improving our paper, as well as the important guiding significance to our researches. We have studied comments carefully and have made correction which we hope meet with approval. Revised portion are marked in yellow in The paper. The main corrections in the paper and the responds to the reviewer’s comments are as flowing:

   Thanks you very much for your comments and suggestions.

Major Comments:

  1. Page 4, distilled water wash (2min x 3times), this sentence need to be rewritten, and in the same paragraph ( dropwise at 37°C30min), add for between C and 30min.

   Answer: We have re-written this part according to your suggestion. And then, we marked the revised content in yellow in the paper.

  1. Page 5, diaminobenzidine (DAB) were added to develop colors under5min at room) replace under with for 5min.

   Answer: Thanks to you for your suggestion. We have made correction according to your comments.

  1. Also, all figure legends were short, more details should be added for all of them.

   Answer: Considering your suggestion, we have added more detailed information for all legends, such as legend titles, methods and results.

   Special thanks to you for your good comments.

                                                      Yours sincerely,

                                                          Honglin QU

Reviewer 3 Report

I suggest you to improve the editing of the text. There are several errors.

Author Response

Dear Reviewers:

   Thank you for your letter and for the reviewers’ comments concerning our manuscript entitled “Aerobic exercise inhibit CUMS-depressed mice hippocampal inflammatory response via activating hippocampal miR-223/TLR4/MyD88-NF-κB pathway” (ID: ijerph-740055). Those comments are all valuable and very helpful for revising and improving our paper, as well as the important guiding significance to our researches. We have studied comments carefully and have made correction which we hope meet with approval. Revised portion are marked in yellow in The paper. The main corrections in the paper and the responds to the reviewer’s comments are as flowing:

   Thanks you very much for your comments and suggestions.

Major Comments:

   I suggest you to improve the editing of the text. There are several errors.

   Answer: We have made correction according to your comments, and we carefully proofread all the content, and indeed found a few undesirable errors.

   Special thanks to you for your good comments.

                                                      Yours sincerely,

                                                          Honglin QU
